# Metaheuristics-Assisted Placement of Omnidirectional Image Sensors for Visually Obstructed Environments

**DOI:** 10.3390/biomimetics10090579

**Published:** 2025-09-02

**Authors:** Fernando Fausto, Gemma Corona, Adrian Gonzalez, Marco Pérez-Cisneros

**Affiliations:** 1Departamento de Innovación Basada en la Información y el Conocimiento, Universidad de Guadalajara, Centro Universitario de Ciencias Exactas e Ingenierías (CUCEI), Boulevard Marcelino García Barragán, No. 1421, Guadalajara 44430, Jalisco, Mexico; 2Departamento de Ingenierías, Universidad de Guadalajara, Centro Universitario de la Costa Sur (CUCSUR), Av. Independencia Nacional, No. 151, Autlán 48900, Jalisco, Mexico; gemma.cnunez@academicos.udg.mx; 3Departamento de Ciencias Básicas, Universidad de Guadalajara, Centro Universitario de la Ciénega (CUCIENEGA), Av. Universidad 1115, Col. Lindavista, Ocotlán 47810, Jalisco, Mexico; adrian.gonzalezb@academicos.udg.mx; 4Departamento de Electro-Fotónica, Universidad de Guadalajara, Centro Universitario de Ciencias Exactas e Ingenierías (CUCEI), Boulevard Marcelino García Barragán, No. 1421, Guadalajara 44430, Jalisco, Mexico; marco.perez@cucei.udg.mx

**Keywords:** metaheuristic optimization algorithms, surveillance, optimal camera placement

## Abstract

Optimal camera placement (OCP) is a crucial task for ensuring adequate surveillance of both indoor and outdoor environments. While several solutions to this problem have been documented in the literature, there are still research gaps related to the maximization of surveillance coverage, particularly in terms of optimal placement of omnidirectional camera (OC) sensors in indoor and partially occluded environments via metaheuristic optimization algorithms (MOAs). In this paper, we present a study centered on several popular MOAs and their application to OCP for OC sensors in indoor environments. For our experiments we considered two experimental layouts consisting of both a deployment area, and visual obstructions, as well as two different omnidirectional camera models. The tested MOAs include popular algorithms such as PSO, GWO, SSO, GSA, SMS, SA, DE, GA, and CMA-ES. Experimental results suggest that the success in MOA-based OCP is strongly tied with the specific search strategy applied by the metaheuristic method, thus making certain approaches preferred over others for this kind of problem.

## 1. Introduction

Biomimetics is an interdisciplinary field devoted to the study and imitation of biological organisms and systems, whose aim is the development of bioinspired solutions to a wide range of complex problems. Biomimetics research has found applications in many fields of science and engineering, including the design of bioinspired mechanisms, sensors and materials, the imitation/modeling of self-organized and cooperative behavior, robotics design, and biomedicine, among others. One of the most important developments that have arisen from biomimetics study are Metaheuristics Optimization Algorithms (MOAs). MOAs (also known as Nature-Inspired Metaheuristics or bio-inspired algorithms) are a family of heuristic computational methods designed to mimic some natural behavior, mechanisms, or laws with the purpose of applying it to solve optimization problems [1]. Many different MOAs have been developed based on an ample variety of phenomena, including natural (Darwinian) evolution (which includes techniques such as Differential Evolution (DE), Genetic Algorithms (GA), Evolution Strategies (ES), etc.), swarm behavior (with methods such as Particle Swarm Optimization (PSO), Grey Wolf Optimizer (GWO), Social Spider Optimization (SSO) and many others) and even the laws of physics (with algorithms such as Gravitational Search Algorithm (GSA), States of Matter Search (SMS) or Simulated Annealing (SA), to name a few). Through the last two decades, MOAs have been successfully applied to solve a plethora of real-world problems from many different areas of application such as engineering design, image processing and computer vision, communications, networks, energy/power management, machine learning, data science, medical diagnosis assistance, robotics, and many others [1].

Video surveillance is a very popular research topic, which is not surprising considering its importance for both security and crime deterring in indoor/outdoor environments. Video surveillance requires the distribution of several image sensors within the target surveillance space (SS). The placement of these devices should be aimed at achieving optimal coverage of the SS; that is, to allow the deployed sensors (camera network) to “observe” as much of the environment as possible at all times [2].

Optimal Camera Placement (OCP) is an optimization problem that consists of finding an optimal number of image sensor and their locations so that they satisfy a maximum coverage of the SS, as well as some other task-specific constraints (i.e., maximum number of available sensors, spatial resolution requirements, area visibility time, etc.). OCP could become a quite challenging task depending on the characteristics of the SS, as well as the number, type and specifications of the available image sensors. Cameras used for surveillance can be classified into three types: 1. Fixed Perspective (FP), 2. Pan-Tilt-Zoom (PTZ), and 3. Omnidirectional. An FP camera consists of an image sensor whose field of view (FoV) models a frustum (pyramid-like shape), and once mounted, they usually remain in a fixed position, orientation, and focal length. FP sensors are often preferred over other camera types due to their low cost and ease of implementation, particularly for indoor surveillance. On the other hand, we have PTZ cameras. Like FP cameras, PTZs FoV models have a frustum shape, whose orientation and focal length can be adjusted thanks to their ability to rotate horizontally (Pan), vertically (Tilt), and by performing optical zoom (up to 40× for most moder cameras). PTZs are often deployed for surveillance in large outdoor environments, such as construction sites or highways, although they can be placed within indoor locations, where security staff can control their movement remotely. While PTZs have useful features, they may have some issues such as command latency (during remote operation), and gaps in security coverage (due to the use of automatic preset motions or when auto tracking errors occur). Finally, there are omnidirectional cameras (OCs), which are mostly used in large interior spaces. The main feature of OCs are their ultra-wide-angle lenses, which can create dynamic viewing angles ranging from 180° to up to 360°. The images produced by OCs are usually circular and with a distorted appearance (which can be corrected by using appropriate de-warping software) and have relatively high resolution (ranging from 2 to 5 megapixel in most commercial models, although higher resolutions are also available). Unlike PTZs, the FoV on OCs often remains fixed to a certain region of interest (ROI). While this may pose a limitation compared to PTZs, this is highly compensated by their wider FoV (which aids in preventing security gaps), and their lack of any mobile parts (eliminating failure due to mechanical wear). In addition, OCs are usually smaller and more discrete compared to PTZs, and thus more difficult to spot by the human eye [2].

Recently, a wide range of solutions for OCP, oriented to specific surveillance tasks, have been proposed. Within these approaches, the use of metaheuristic optimization algorithms (MOA) for OCP problem is of special interest. For example, in [3] FP cameras are randomly scattered within a ROI, and then, Particle Swarm Optimization (PSO) is deployed to find a set of optimal sensor orientations so that the camera network’s coverage is maximized. Such approach was compared against that of Potential field-based Coverage Enhancing Algorithm (PFCEA), achieving better results in terms of coverage. Also, in [4] a variant of a Binary Particle Swarm Optimization (BPSO) known as BPSO Inspired Probability (BPSO-IP) was proposed for the automated placement of FP camera networks, with emphasis on both locations, and orientations. In this study the authors compared the performance of BPSO-IP against other well-known BPSO-based techniques, as well as with methods such as binary GA (BGA), Simulated Annealing (SA) and Tabu Search (TB), achieving good results in terms of both deployment cost and coverage. In [5] an approach for OCP based on a variant of SA known as Trans-dimensional Simulated Annealing (TDSA) was presented and compared against camera placement methods based on Binary Integer Programming (BIP) and two other heuristics. In this study, both omnidirectional and FP cameras were considered for experimentation, and results show that TDSA can deliver placement configurations which maximize coverage with a more reduced number of sensors, making it more attractive for application compared to the other methods. In [6] a variant of GA called Guided Genetic Algorithms (GGA) is applied for OCP in multi-camera motion capture systems. In this approach, a distribution/estimation technique is first applied to restrict the search space prior to GGA initialization. Furthermore, an error metric (optimization function) is also proposed to evaluate the quality of sensor placement in the camera network. This approach was compared to that based on the traditional GA, demonstrating a better performance. In [7] an approach for the minimization of total cost in camera placement for bridges’ surveillance was presented. Here the authors propose to apply a greedy algorithm and GA, along with a visibility test as an approach to exploring possible solutions effectively. In addition, a heuristic called Uniqueness score with Local Search Algorithm (ULA) was developed and compared against the greedy algorithm and GA, outperforming them in terms of average cost, coverage, and computation time. In [8] an OCP approach for the surveillance of indoor work areas based on Grey Wolf Optimizer (GWO) was proposed. In this approach, the objective was to set three PTZ cameras within a certain SS (priority area) with the aim to maximize information content and minimize the effects of occlusions caused by randomly moving objects. This approach was compared against methods based in GA and PSO, demonstrating to be preferred to both in terms of computational complexity and convergence speed. In [9] an improved GA was proposed for OCP in metro station construction sites with the aim of setting a minimum number of PTZ sensors while also achieving total coverage of the construction and tackling the problem of occlusions caused by various dynamic elements present during construction.

The current literature suggests that there is a significant amount of research dedicated to MOA-based OCP for either FP or PTZ cameras; in contrast, the literature seems to be somewhat lacking in works related to optimal placement of OCs assisted by MOAs. In attention to this research gap, in this paper, we present a comparative study of several popular MOAs applied to OCP of OCs in indoor environments. The aim of this comparison is to observe the performance of different MOAs in several OCP tasks. For our experiments, we first model the OCP problem in terms of sensor characteristics (particularly FoV) and target SS (including sources of occlusions such as walls or columns), and then, MOAs are applied to deliver an optimal set of locations for prespecified amounts of image sensors. Our comparisons include camera placement configurations delivered by Particle Swarm Optimization (PSO) [10,11], Grey Wolf Optimizer (GWO) [12,13], Social Spider Optimization (SSO) [14,15], Gravitational Search Algorithm (GSA) [16,17], States of Matter Search (SMS) [18], Simulated Annealing (SA) [19], Differential Evolution (DE) [20,21], Genetic Algorithms (GA) [22,23] and Covariance Matrix Adaptation Evolution Strategies (CMA-ES) [24].

The rest of this paper is organized as follows: In Section 2, we present the problem formulation for OCP. In Section 3, a review about metaheuristic optimization algorithms emphasizing their main attributes is presented. In Section 4, we present our experimental results. Finally, in Section 5, conclusions are drawn.

## 2. Problem Formulation

The main objective in OCP is to maximize the amount of area observed by a camera network (set of image sensors) within a target surveillance space (SS). For a 2-D SS, the placement configuration of the camera network may be described expressed as $s=\left[x_{1},y_{1},x_{2},y_{2},\ldots ,x_{N_{sensors}\;},y_{N_{sensors}\;}\right]$, where each pair $\left[x_{i},y_{i}\right]\;=\;z_{i}$ denotes the location of the j-th camera within the 2-D space, while $N_{sensors}$ stands for the number of available image sensors. Since the amount of area observed by the camera network is relative to both, the structure of the SS and the location of the visual sensors themselves, the optimization problem to solve may be expressed as follows:(1)$$Maximize:\;\;\;F\left(s\right)\;=\;\frac{A\left(P\left(s\right)\right)}{A\left(P_{M}\setminus {⋃}_{k=1}^{K}h_{k}\right)}\;\;\;Subject\;to:\;\;\;x_{j},y_{j}\in P_{M}$$
where $A\left(\cdot \right)$ is the area operator. Furthermore $P_{M}$ denotes de main polygon describing the target SS (deployment area), while $h_{k}$ $\left(k\;=\;1,\;2,\ldots ,K\right)$ denotes simple holes (polygons within the SS representing potential sources of visual obstruction, such as walls, columns, furniture, etc.). Also, P(s) is the polygon representing the area of the surveillance space that is observed by the camera network, as given by(2)$$P\left(s\right)={⋃}_{j=1}^{N}\;p(z_{j},\pi )$$
where $p\left(z_{j},\pi \right)$ is the polygon representing the area within the SS that is observed by the j-th camera according to its respective location $z_{j}=\left[x_{j},y_{j}\right]$ and sensor parameters/restrictions π, as given by(3)$$p\left(z,\pi \right)=v(z)\cap c(z,\pi )$$
with v(z) denoting the visibility polygon of the visual sensor (extension of the SS that is visible from a certain location when assuming a full 360° field of view, unlimited spatial resolution and infinite depth of field) computed at point z, whereas c(z,π) stands for the polygon describing the default field of view (FoV) of a camera sensor placed at point z and regarding its corresponding set of characteristics π, which may include properties such as orientation (as in the case of Fixed or PTZ cameras) or minimum required resolution (which may be relevant for tasks such as detection and/or recognition). For example, if we consider a PTZ camera placed for human recognition purposes within a 2-D SS, its default FoV will be determined by the orientation angle ψ (azimuth) of its viewing frustum at point z, as well as the maximum distance D at which the minimum resolution required for recognition is satisfied (see Figure 1a). On the other hand, if an omnidirectional visual sensor is considered instead, then its default FoV will be given as a circular region of radius D centered a point z (see Figure 1b).

As for the procedure for determining the visibility polygon v(z), it can be formulated as follows: Given the simple polygon $P_{M}$, a set of K simple holes $H\;=\;\left\{h_{1},h_{2},\ldots ,h_{K}\right\}$ and the camera placement point z = [x,y] such that(4)$$h_{k}\subset P_{M}\;\;\forall k$$
and $\partial h_{i}\cap \partial h_{j}\;=\;\emptyset :i\neq j\;\forall i\;\forall j$ (with ‘∂’ denoting the boundary operator)(5)$$z\in P_{M}\wedge z\notin h_{i}\;\forall i$$

The visibility polygon v(z) will be comprised by the set of all points p within $P_{M}$ such that(6)$$v\left(z\right)\;≜\;\left\{p:p\in P_{M}\;\wedge p\notin h_{i}\forall i\wedge \underline{zp}\subset \left\{P_{M}\setminus {⋃}_{k=1}^{K}h_{k}\right\}\right\}\;$$

To compute the visibility polygon at point z, we can apply the algorithm proposed in [1]. For a simple polygon $P_{M}$ without holes $h_{i}$, such methodology consists of performing a counterclockwise (CCW) radial sweep of the polygon over the range [0,2π] with point z as the center. The purpose of this is to compute the visible line segments emanating from z and then represent v(z) as the union of all such segments. To apply this procedure, it is first required to represent the polygon $P_{M}$ (deployment area) as an edge list in cartesian coordinates as follows:(7)$$ELC\;≜\;\left\{\left({vc}_{1}^{s},{vc}_{1}^{e}\right),\left({vc}_{2}^{s},{vc}_{2}^{e}\right),\ldots ,\left({vc}_{i}^{s},{vc}_{i}^{e}\right),\ldots ,\left({vc}_{E}^{s},{vc}_{E}^{e}\right)\right\}$$
where E is the number of edges, i the edge index ordered CCW, $\left({vc}_{i}^{s},{vc}_{i}^{e}\right)\in R^{2}$ are the start (s) and end (e) vertices of edge ‘i’ and ${vc}_{i}^{e}\;=\;{vc}_{i+1}^{s}$.

Once the cartesian edge list ELC is defined, it is then converted to its polar coordinate representation; this is(8)$$ELP\;≜\;\left\{\left({vp}_{1}^{s},{vp}_{1}^{e}\right),\left({vp}_{2}^{s},{vp}_{2}^{e}\right),\ldots ,\left({vp}_{i}^{s},{vp}_{i}^{e}\right),\ldots ,\left({vp}_{E}^{s},{vp}_{E}^{e}\right)\right\}$$
where ${vp}_{i}≜\left\{\theta_{i},r_{i}\right\}$ are the polar angle and radius of the i-th edge’s vertex, respectively.

With ELP now defined, the algorithm then proceeds to discard all edges that might not be visible (i.e., those for which $\theta^{e}\leq \theta^{s}$) and then the remaining vertices are sorted in lexicographical ascending list as follows:(9)$$Q\;≜\;\left\{\left(\theta_{1}^{s},r_{1}^{s},\epsilon_{1}\right),\ldots ,\left(\theta_{E}^{s},r_{E}^{s},\epsilon_{E}\right),\left(\theta_{1}^{e},r_{1}^{e},\epsilon_{1}\right),\ldots ,\left(\theta_{E}^{e},r_{E}^{e},\epsilon_{E}\right)\right\}$$
where θ, r and ϵ are the vertex, polar angles, radii, and edge pointers, respectively. The resulting vertex list Q is finally applied by the algorithm to sweep the polygon in CCW order, outputting the visibility polygon v(z).

The extension of this algorithm for a polygon $P_{M}$ with polygonal holes $h_{k}$ is straightforward. Since the outside of holes are also well defined by their respective edge lists (in a clockwise order,) it is only required to append such lists to $P_{M}$’s edge list, and then apply the algorithm as described above [1] (see Figure 2).

## 3. Metaheuristic Optimization Algorithms

Metaheuristic optimization algorithms (MOAs) have become quite popular during the last decades due to their diversity in terms of design and applications; they have been widely applied to solve problems competing to different areas of interest, such as engineering design, communications, energy management, data analysis, computer vision, robotics, among others [2]. MOAs are often modeled as population-based optimization strategies, in which a set of search agents representing candidate solutions are constantly updated by following certain rules until a satisfactory result can be delivered [25,26]. Most of these optimization approaches are composed of three main steps: 1. initialization, 2. main loop, 3. solution presentation. For the initialization step, it is common for MOAs to start by generating a population of random individuals $X\;=\;\left\{x_{1},x_{2},\ldots ,x_{N}\right\}$ (where N is the population size), where each of its elements $x_{i}$ denotes a candidate solution for a given optimization problem. As for the main loop, this is where similarities among MOAs start to diverge; it is during this step where a search strategy, characteristic of each algorithm, is applied to iteratively update the population of solutions, aiming to improve their quality. Once the main loop comes to an end (due to meeting a pre-specified stop criterion), the algorithm enters its final step, in which it chooses the best solution found during the search process and reports it as the best approximation to the global optimum [27].

Search strategies applied by MOAs may be quite different from one algorithm to another, which of course has an impact on the performance of these methods against certain optimization problems. In MOAs such as DE or GA [22,28], for example, solutions may be generated by either “mixing” information from randomly chosen solutions (crossover) or by adding some perturbations to currently existing ones (mutation) [21,29,30]. On the other hand, there are techniques in which new solutions are produced by accounting for a series of “attraction” effects among certain individuals in the population. Such are the cases of algorithms like PSO [10,11,31], ABC [32,33] or GSA [16,34], where solutions are set to move toward seemingly good solutions among the present population and/or the current global best solution found by the search process. Another trait impacting the performance of MOAs’ search strategies is their population selection mechanism; in this sense, MOAs can be classified as either greedy or non-greedy. In the case of greedy algorithms (such as GA) the population of solutions for the next iteration of the search process is set by selecting the best elements among the current population and a set of newly generated candidate solutions, then discarding all remaining elements. This type of selection mechanism allows a fast convergence toward promising solutions, but usually with the cost of compromising exploring capabilities [2,35]. A variation in this mechanism (individual greedy) can be found in algorithms such as DE or ABC, where a newly generated solution is accepted only if it directly improves the solution from which it was generated. In contrast to typical greedy selection, this approach has the appeal of being more balanced in terms of exploration–exploitation ratio, as the elements of the initial population are forced to improve individually from their own starting point [2,35]. On the other hand, non-greedy algorithms (such as PSO, GSA or GWO) have the distinction of not considering the relative quality of newly generated solutions, thus accepting all of them (independent of their quality) as replacement for the current population at the end of each iteration. The lack of elitism in non-greedy selection is preferred when exploration is preferred over exploitation [2,35]. It is worth nothing that the performance of a MOA cannot be predicted by only analyzing the characteristics of its applied search strategy, but rather by how these strategies interact with a specific problem’s features (such as the number of decision variables, bounds, constraints, modality, objective(s), etc.). From a practical point of view, this suggest that there will be certain search strategies that may perform better than others across certain problems, whereas for others they might be outperformed by another algorithm [35,36]; this fact has served as an inspiration for researchers to keep developing new and more efficient search strategies, aimed at offering better solutions to an ever-increasing variety of problems. Techniques worth mentioning aimed at this continuous search for improvement include modifications/hybridizations of popular MOAs such as PSO (with variants including those which combine crossover [37] and mutation [38] operations, and hybridizations such as Clustering-Based Hybrid Particle Swarm Optimization (CBHPSO) [39], Real-Coded Genetic Algorithms with PSO (RCGA-PSO) [40]), GA variants (with examples such as with Ant Colony Optimization (GA-ACO) [41] and the Grey Wolf Optimizer/Genetic based optimizer (GWO-GA) [42]), and many others.

## 4. Experimental Results

Our proposed MOA-based OCP Sensor Deployment Scheme consists of implementing MOAs to optimize the locations configuration of a set of OCs so that the coverage area rate (the amount of area that is observed by the sensor network) is maximized. In this approach, the function F(s) presented in Equation (2) is applied as the fitness function applied by each MOA as part of their search process (see Figure 3). Here it is important to remember that $s=\;\left[x_{1},y_{1},x_{2},y_{2},\ldots ,x_{N_{sensors}\;},y_{N_{sensors}\;}\right]$ is comprised by xy pairs of sensor locations (i.e., [$x_{i},y_{i}$] being the xy location associated with the i-th image sensor); in order to match this with the representation of search agents in MOAs, let $p=\;\left[p_{1},p_{2},p_{3},p_{4}\ldots ,p_{N_{dims}}\right]$ represent a single search agents (with $N_{dims}$ being the number of decision variables in a given optimization problem). The matching between the elements of s (xy locations) and p is as given as follows:(10)$$\left[x_{1},y_{1},x_{2},y_{2},\ldots ,x_{N_{sensors}\;},y_{N_{sensors}\;}\right]\;=\;\left[p_{1},p_{2},p_{3},p_{4}\ldots ,p_{N_{dims}-1},p_{N_{dims}}\right]$$This means that for each xy pair $[x_{i},y_{i}]\in s$ there will be a corresponding pair of successive elements $[p_{j},p_{j+1}]\in p$. In order to match each xy pair $\left[x_{i},y_{i}\right]$ with a pair of elements from p, the number of elements $N_{dims}$ represented by each search agent should be twice the number of available image sensors $N_{sensors}$; that is(11)$$N_{dims}=N_{sensors}\times 2$$

In other words, the dimensionality (number of decision variables) of the optimization problem (maximization of coverage area rate) is conditioned by the number of available image sensors (the greater the number of sensors, the higher the dimensionality).

As for our experiments, we have considered two different deployment layouts (see Table 1). The first layout ($L_{1}$) consists of a square-shaped deployment area (polygon) with a size of 100 m × 100 m and a rectangular hole (representing a visual occlusion) in the middle. The second layout ($L_{2}$) is represented by a rectangular polygon with a size of 100 m × 60 m, containing two holes (a square-shaped polygon and a J-shaped polygon). The region that is required to be covered by the camera network for both $L_{1}$ and $L_{2}$ is represented by the gray-colored background in each layout, and their area sizes are 9200 m^2^ and 4085.5 m^2^, respectively. The image sensors considered in our experiments are divided into two types: the first type ($C_{1}$) is a simple OC with horizontal resolution of 1280 px (a common resolution for most “economic” cameras). The second type of sensor ($C_{2}$) is another omnidirectional camera with horizontal resolution of 3584 px (common for “premium” image sensor). Since both cameras are meant to be applied for human recognition in indoor spaces, it is of special interest to know the maximum distance for recognition (minimum resolution distance) associated with each sensor; in the case of image sensor $C_{1}$ such distance is of up to 10.24 m, whereas for sensor $C_{2}$ it is up to 28.67 m (see Table 2).

By considering the proposed deployment layouts and camera models, we can build four different experimental setups, each consisting of a layout (either $L_{1}$ or $L_{2}$), a camera model ($C_{1}$ or $C_{2}$), and a maximum number of sensors $N_{sensors}$ (see Table 3). Since each camera model has a different maximum recognition distance (and thus a different size on their FoV), the maximum number of sensors required to maximize the coverage of a given deployment space will also be different. To estimate the appropriate number of cameras $N_{sensors}$ to apply at each experimental setup, we simulated a manual colocation of image sensors over the deployment layout seeking maximum coverage and minimum overlap of the sensors’ FoV.

In our experiments, nine different MOAs were tested: PSO, GWO, SSO, GSA, SMS, SA, DE, GA, and CMA-ES. Each of these algorithms proposes a different search strategy, whose performance can be slightly modified by changing the setup of their distinctive algorithm parameters; more importantly, these algorithms were chosen with the intention of ensuring variety with respect to certain performance-influencing characteristics: 1. selection mechanisms, 2. type of attractors (when applied), 3. dependence on iterative process, 4. use of population sorting mechanism, 5. use of measurements related to population/agents, and 6. additional memory requirements. A summary of these characteristics is presented in Table 4.

In preparation for our experiments, an exhaustive tuning process was applied to find the best parameter configuration for each of the chosen MOAs. Such setups are described in Table 5.

As for the test settings, to ensure a fair comparison between all tested MOAs, population size for each algorithm was set to $N_{pop}\;=\;50$, while the stop criterion for each of these is dictated by the maximum number of allowed fitness function evaluations (function accesses) which was set as $N_{FA}\;=\;5000$. The remaining test settings (number of decision variables and search space bounds) are strictly dependent on each of the proposed experimental setups (Setup1, Setup 2, Setup 3, and Setup 4, as described in Table 3); specifically, the number of decision variables $N_{dims}$ is set to be twice the maximum number of available image sensors for each setup. This is because the solution vector is coded to represent the xy coordinates $\left[x_{i},y_{i}\right]$ corresponding to each image sensor i, as described in Section 3. As for the bounds of the search space, these are dictated by the size of the deployment area (layout) modeled on each experimental setup, i.e., for Setup 3, the lower bounds lb will be represented as the origin location ($lb\;=\;\left[0,\;0\right]$), whereas the upper bounds ub are denoted by the maximum horizontal and vertical length of the deployment area ($ub\;=\;\left[100,\;60\right]$ in this case). The settings for the number of decision variables and the lower/upper bounds corresponding to each experimental setup are summarized in Table 6.

All chosen MOAs were applied over each of the proposed experimental setups. For each case, each MOA was tested a total of 30 times, and the results (best fitness value delivered by the search process) have been registered and reported in terms of five performance indexes: 1. the mean for the best fitness values found for each set of 30 individual tests ($f_{mean}$), 2. the median of said fitness values ($f_{median}$), 3. their standard deviation ($f_{std}$), 4. the lowest fitness value found among each set of tests ($f_{worst}$), and 5. the highest fitness value delivered by each set of tests ($f_{best}$). The performance results delivered by all tested MOAs over each experimental setup are reported on Table 7, Table 8, Table 9 and Table 10, where the best values for each performance index is shown in boldface. Also, the rank assigned to each algorithm according to its performance over each setup is shown at the bottom of each of these tables.

For Setup 1, we can see that the best performing algorithm is GA, followed by CMA-ES and SSO. On the other hand, for Setup 2, the best performing algorithms are represented by CMA-ES, GA and GWO. For Setup 3, GA is once again the best ranked algorithm, followed by SSO and GWO. Finally, for Setup 4, PSO has the best ranking, being followed by GA and SSO.

Furthermore, Figure 4 presents the fitness evolution curves showing the average performance of each of the applied MOAs for each of the proposed experimental setups. These plots yet again give insight into the great performance of algorithms such as GA, CMA-ES, SSO, and PSO which have the ability reach significantly high fitness values by the end of the search process; however, these plots also expose the underperformance of algorithms such as DE and GSA, which suffer from stagnation very early in their respective search processes, or SMS, which is unable to deliver any competent solution for the proposed experimental setups. Figure 5, Figure 6, Figure 7 and Figure 8 show the best image sensor placement considering each experimental setup and the tested MOAs. Finally, in Table 11 we present the overall ranking assigned to each algorithm according to its performance over the whole set of experimental setups. In general, the best performing algorithm is represented by GA (which is not surprising, considering its prevalence among the top three algorithms across all experimental setups), followed by PSO and SSO. On the other hand, the worst performing algorithms are represented by SMS, SA, and GSA, with each of them consistently attaining low rankings in all experimental setups.

In addition to performance tests, processing times associated with each of the test MOAs when applied to each experimental setup has also been documented. The average processing time (in seconds) for each tested MOA/experimental setup is shown in Table 12 (numerically) and Figure 9 (graphically).

## 5. Discussion

As previously illustrated in Table 11, the best overall performing algorithm for the proposed OCP problems is GA. The GA method is an evolutionary algorithm (classification which also groups algorithms such as DE and CMA-ES) [2]. Evolutionary algorithms (EAs) distinguish themselves among other MOAs due to the use of operators which simulate phenomena observed in natural evolution, such as crossover, mutation, and selection. The way in which these operators are modeled and applied is usually different among EAs. In the case of GA, for example, the objective of these operators is to generate a new set of solutions by applying both crossover and mutation operators to the currently existing population, and then comparing such new individuals with the previously existing ones in terms of fitness to choose the best individuals among these two groups to define the new population for the next generation (this is called a greedy selection criterion) [22]. On the other hand, in the case of DE, mutation operators are applied first to generate a new candidate (mutant) solution from each member in the population, and then crossover operations are performed to mix the information of each current solution and its corresponding mutant solution, forming a set of trial solutions. Finally, trial solutions are compared against their corresponding originating individual, and then the best among each pair is selected and kept as part of the population for the next generation, while the others are discarded (this represents an individual greedy selection criterion) [29]. Finally, for CMA-ES (a variant of ES), while crossover and mutation operators are applied (according to the algorithms’ own approach), specific selection operators are absent, thus accepting any new solution generated by the crossover/mutation operators independent of their quality (non-greedy selection mechanism) [36]. From a selection mechanism perspective, it seems that the greedy selection criterion associated with GA offers an advantage when applied to solve OCP problems in contrast to non-greedy methods (as in the case of CMA-ES). The good performance of GA for OCP may also be related to the nature of its own crossover and mutation operators. In contrast to DE and CMA-ES, the probabilistic criterion applied by GA to select parent solutions and the way recombination of information is applied in its crossover seems to be more effective in promoting solution diversity, thus allowing better exploration of the feasible solution space; for OCP problems, this translates into an ability for finding better placement distributions for the available image sensors, and thus better coverage of the target surveillance space.

The second and third best MOAs for OCP according to our experiments are PSO and SSO. It is interesting that these two methods (both classified as swarm-based algorithms) search operators which model attraction toward some previously defined location(s) or individuals. In the case of PSO, for example, the attraction movements performed by each particle in the population is dictated by both, the location of the global best solution (stored in the algorithms “memory” at the current iteration) and its personal best solution. Modeling the attraction movements in terms of these two specific solutions allows search agents not only to converge toward the location of the global best solution, but also to move toward the direction of other prominent solutions, which enhance their exploration capabilities [10,11]. Similarly, in SSO, movements are modeled so that search agents move toward other prominent members of the population. The key difference with PSO though, is that SSO models a population composed of two types of search agents: female and male individuals (the latter of which are further classified as either dominant or non-dominant according to their aptitudes). Under this scheme, SSO can deploy different movement strategies associated with both their gender designation and how each of them can interact with other agents (according to their specific movement rules), thus giving it flexibility for exploring the feasible search space. Also, SSO proposes a mating and selection mechanism similar to crossover operators present in EAs which allows it to generate new candidate solutions by mixing information corresponding to female and (dominant) male individual, and then comparing them to some of its originating solutions to select the best among them (thus performing a sort of greedy selection) [15]. While both PSO and SSO share some similarities in terms of search strategy, the clearest difference lies in the way the movement of search agents is guided; in the case of PSO, it uses “memory” to store and update the information of prominent solution found during the search process (namely, the global best solution and the personal best solutions associated with each particle) and can use this information to guide the movement of the population toward those prominent locations. Conversely, SSO does not implement a process to register prominent solutions at all, and instead updates the population by using the information provided at the moment by those same individuals (which could cause search agents to diverge away from the global optimum depending on how the search process has developed). Finally, it is worth discussing a key difference between GA (the best-ranked algorithm) and PSO (the second best). As previously discussed, in GA new solutions are generated by applying both crossover and mutation operators, followed by a selection process in which old and new solutions are compared against each other to keep the most prominent among them. In contrast, PSO applies attraction-based operators which drive the population toward the location of the currently known global and personal best solutions and to update accordingly. The differentiating factor here is the presence of an actual (greedy) selection process in the case of GA. In contrast, while PSO solutions are driven toward selected locations of the search space at each iteration, these are always updated independently of their quality compared to their previous locations (thus being classified as a non-greedy algorithm).

In summary, for problems like OCP, in which the distribution of image sensors must be carefully defined to ensure maximum coverage rate, it seems that the unsupervised nature of a non-greedy selection algorithms makes better solutions harder to deliver in contrast to those based on Greedy selection schemes. This is evidenced by the performance shown by algorithms such as GSA, SMS, GWO, and SA during our experiments, which are, very appropriately, non-greedy algorithms. While PSO and SSO also fall in this category, the inclusion of memory for storing the location of prominent solutions, along with the nature of its search strategy, compensates for its lack of elitism, thus allowing them to deliver good results when applied of OCP of OC-type sensors.

As for the computing time tests, it can be seen from Table 12/Figure 9 that processing time (in seconds) for all tested MOAs is drastically higher in Setups 1 and 3 (compared to Setups 2 and 4); this makes sense, since for such experimental configurations, the number of deployed sensors is significantly greater (34 and 20 vs. 6 and 5, respectively). Here, it is important to remember that the dimensionally (number of decision variables) of the optimization problem to solve is conditioned by the number of sensors that are required to be allocated (twice the number of image sensors, as shown in Equation (11)); thus, the higher the number of available sensors, the more complex the optimization of their arrangement. In addition, it is worth mentioning that most of the processing time for a single optimization test is associated with the evaluation of the fitness function itself; this is because calculating coverage area rate under the proposed radial sweep approach involves a procedure in which individual steps are complex (and slow) to compute (i.e., computing each sensor’s visibility polygon and visible area, followed by the calculation of the total surveillance area and the coverage rate in terms of the total deployment polygon’s area). This, added to the complexity of optimizing numerous sensors, makes the whole process relatively slow, taking at least 131 s (approximately 2.2 min) even in the fastest documented test (CMA-ES when tested for Setup 2). As for the MOAs tested, a tendency toward an increased computing time can be observed in algorithms such as SSO and PSO (both swarm algorithms with attractor-based operators), as well as in GA. The lengthier computing times in the case of SSO (averaged ranking as first in this category) may be related to the varied and more complex operators used to update solutions, as well as the different designations adopted by search agents within the algorithms (male/female individuals, as well as dominant/non-dominant members, all of which require the use of additional memory) plus the integration of mating-like mechanism (which compared to most other MOAs, could demand additional fitness function accesses within a single iteration of the process). In the case of PSO, the computing time (on average ranked as the second highest) this could be attributed to the use of memory mechanisms. These mechanisms store local and global promising solutions, which are then used in the calculation of the updated population. In contrast, the lengthier computing times in the case of SSO (on average ranked first in this category) may be related to the varied and more complex operators used to update solutions, as well as the different designations adopted by search agents within the algorithms (male and female, as well as dominant and non-dominant individuals) plus the integration of mating-like mechanism (which, compared to most other MOAs, could demand additional fitness function accesses within a single iteration of the process). Finally, there is the case of GA (ranking third in average highest computing time) which in contrast to SSO and PSO is an evolutionary algorithm. In this case, the increased computing time might be associated with the crossover, mutation, and selection operators (and their respective related parameter configurations) applied as part of their iterative process; in particular, since the mutation rate in our tested GA is set to a relatively high value ($c_{p}\;=\;0.7$) the algorithm is more prone to alter elements in the available search agents, thus increasing the number of operations performed.

## 6. Conclusions

In this paper, different MOAs have been tested for the task OCP of OC-type sensors in visually obstructed indoor environments. Experiments which simulate the placement of several OCs in environments with different characteristics where performed. MOAs tested in our experiments include PSO, GWO, SSO, GSA, SMS, SA, DE, GA, and CMA-ES. As shown in our experiments, GA stands as the best performing technique overall; the reason for its success is related to both its simple but effective crossover/mutation mechanisms, as well as its elitist selection scheme, which results in a heightened exploitation of promising solutions. The other two algorithms that excelled in our tests are PSO and SSO. These MOAs are different to GA as they update solutions by relying on attractor-type operators rather than crossover and/or mutation. However, different to other attraction-based MOAs applied in our tests, PSO and SSO have several advantages which help them to handle OCP better than the other techniques such as the use of memory for storing information on relevant solutions found during the search process which compensates for their lack of elitism in contrast to GA. In conclusion, MOAs that either follow greedy-selection criteria or use memory to keep track of potentially relevant solutions found during the search process seem to be the best choices to tackle OCP problems where environmental obstructions are present within the surveillance space.

## 7. Future Work

The selection of MOAs considered in our experiments was performed considering their simplicity (in terms of search strategy and parameter tuning), as well as several performance-influencing characteristics (selection mechanism, type of attractors, iterative process dependence, etc.), as it was of particular interest to analyze how these collections of traits impact the performance of MOAs over the studied OCP problem. Future work in this regard includes testing optimization techniques which propose more sophisticated search strategies, including modifications and hybridizations of well-known MOAs such as PSO, GA, DE, etc., as well as other mainstream algorithms currently reported in the literature.

## Figures and Tables

**Figure 1 biomimetics-10-00579-f001:**
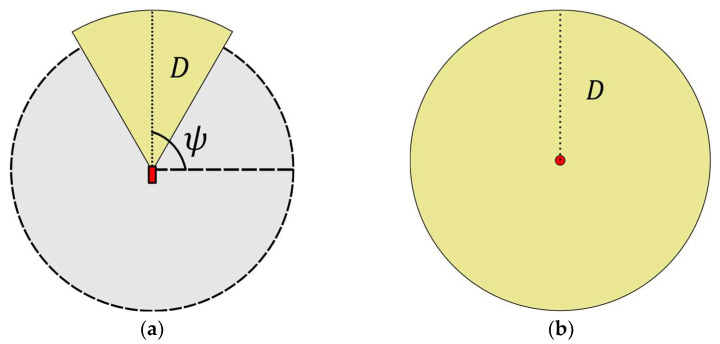
Field of view (FoV) representation for two camera model types placed in 2-D space: (**a**) PTZ camera’s FoV showing its orientation angle ψ, and minimum resolution distance D; (**b**) omnidirectional camera’s FoV delimited by its minimum resolution distance D.

**Figure 2 biomimetics-10-00579-f002:**
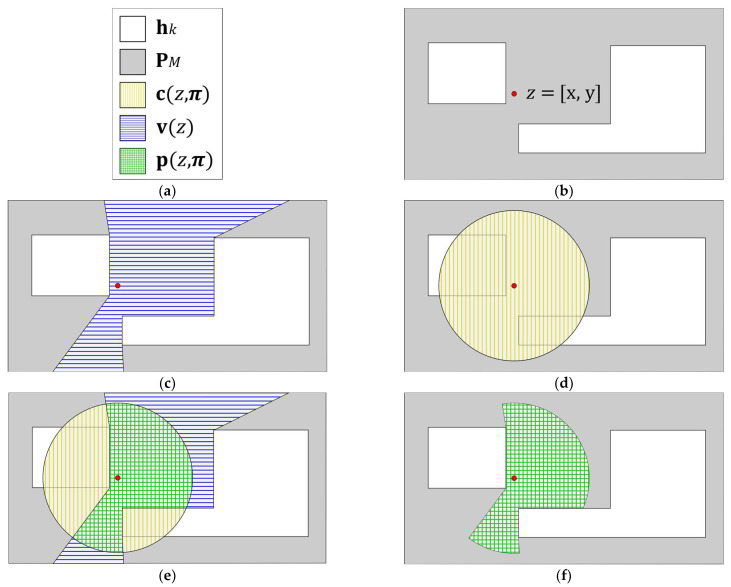
Delimitation of a visual sensor’s observed area: (**a**) symbology; (**b**) surveillance space layout showing its main polygon $P_{M}$ (deployment area), its holes $h_{k}$, and a placement point z = [x,y]; (**c**) visibility polygon v(z) for a visual sensor placed in point z; (**d**) visual sensor’s default field of view c(z,π) corresponding to an omnidirectional camera; (**e**) intersection of the visual sensor’s visibility polygon v(z) and default field of view c(z,π); (**f**) area of the surveillance space that is observed by the image sensor placed in point z.

**Figure 3 biomimetics-10-00579-f003:**
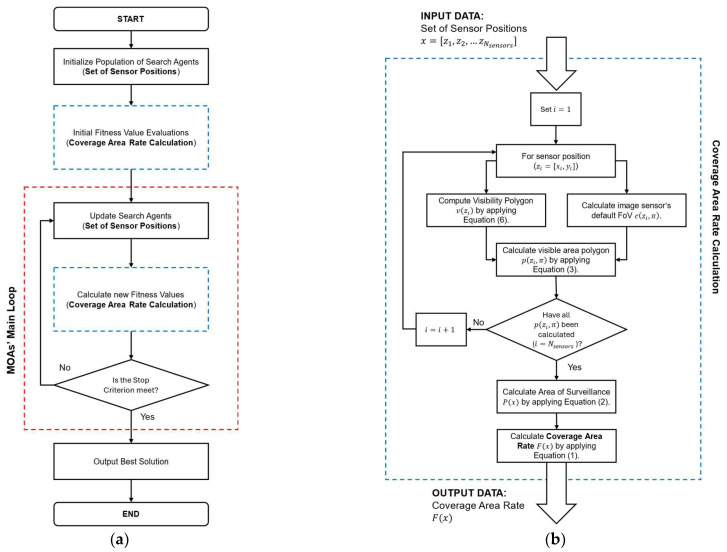
Flowchart of the proposed MOA-based OCP Sensor Deployment Scheme: (**a**) main flowchart highlighting MOAs’ main loop (red dashed box) and coverage area rate calculation (blue dashed box); (**b**) flowchart illustrating the coverage area rate (fitness function) calculation procedure.

**Figure 4 biomimetics-10-00579-f004:**
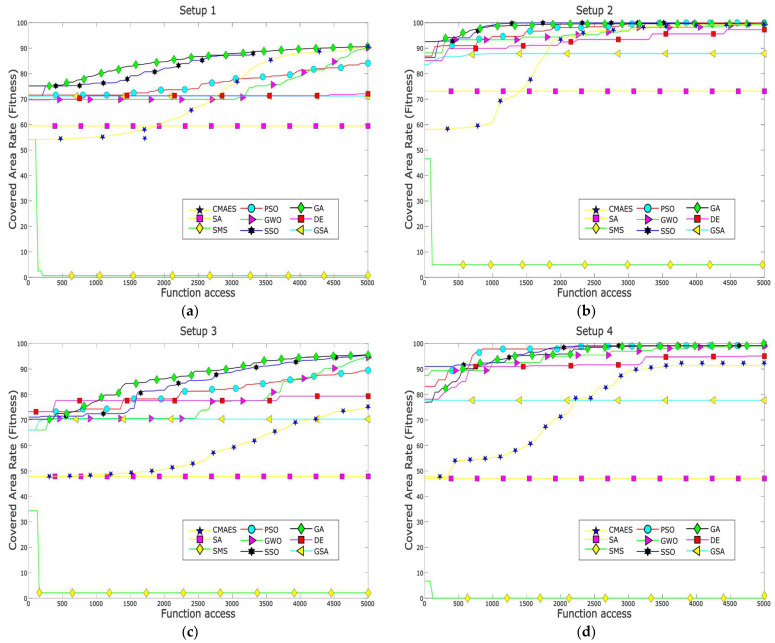
Fitness evolution curves of all tested MOAs across each test setup. Each curve corresponds to the averaged performance (average best fitness values) over 30 individual test runs per MOA: (**a**) Setup 1; (**b**) Setup 2; (**c**) Setup 3; (**d**) Setup 4.

**Figure 5 biomimetics-10-00579-f005:**
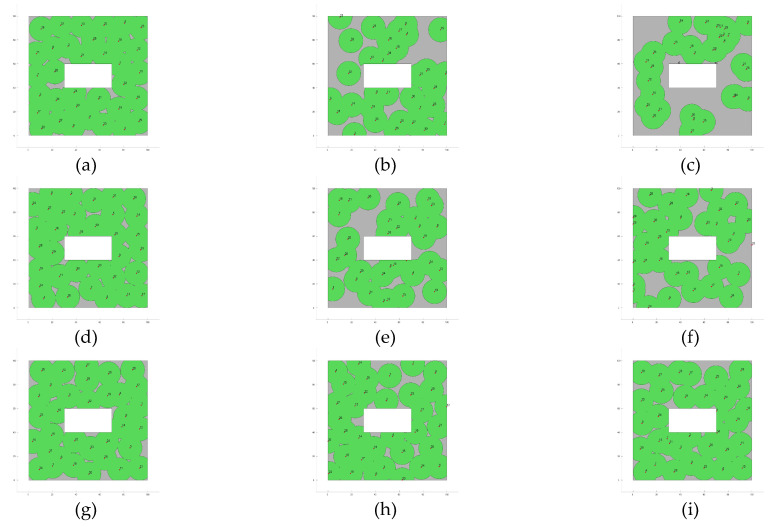
Best image sensor placements for Experimental Setup 1 (Layout $L_{1}$, camera model $C_{1}$ and max. number of sensors $N_{sensor}\;=\;34$). The green circles show the area covered by each deployed sensor. Tested MOAs include (**a**) CMA-ES; (**b**) SA; (**c**) SMS; (**d**) GA; (**e**) GSA; (**f**) DE; (**g**) GWO; (**h**) PSO; (**i**) SSO.

**Figure 6 biomimetics-10-00579-f006:**
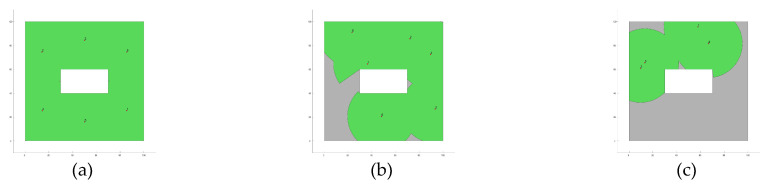
Best image sensor placements for Experimental Setup 2 (Layout $L_{1}$, camera model $C_{2}$ and max. number of sensors $N_{sensor}\;=\;6$). The green circles show the area covered by each deployed sensor. Tested MOAs include (**a**) CMA-ES; (**b**) SA; (**c**) SMS; (**d**) GA; (**e**) GSA; (**f**) DE; (**g**) GWO; (**h**) PSO; (**i**) SSO.

**Figure 7 biomimetics-10-00579-f007:**
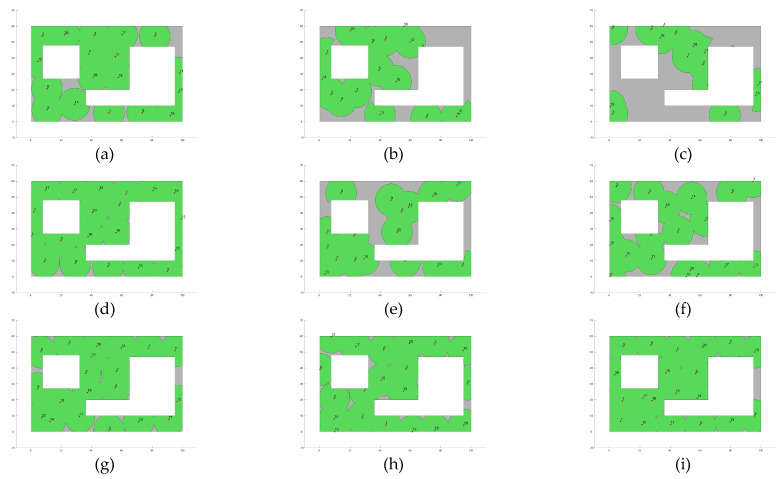
Best image sensor placements for Experimental Setup 3 (Layout $L_{2}$, camera model $C_{1}$ and max. number of sensors $N_{sensor}\;=\;20$). The green circles show the area covered by each deployed sensor. Tested MOAs include (**a**) CMA-ES; (**b**) SA; (**c**) SMS; (**d**) GA; (**e**) GSA; (**f**) DE; (**g**) GWO; (**h**) PSO; (**i**) SSO.

**Figure 8 biomimetics-10-00579-f008:**
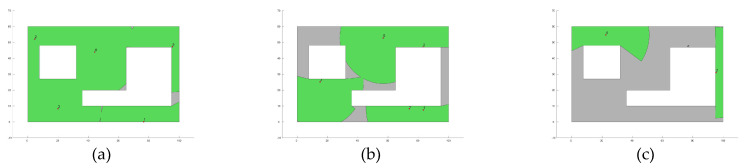
Best image sensor placements for Experimental Setup 4 (Layout $L_{2}$, camera model $C_{3}$ and max. number of sensors $N_{sensor}\;=\;5$). The green circles show the area covered by each deployed sensor. Tested MOAs include (**a**) CMA-ES; (**b**) SA; (**c**) SMS; (**d**) GA; (**e**) GSA; (**f**) DE; (**g**) GWO; (**h**) PSO; (**i**) SSO.

**Figure 9 biomimetics-10-00579-f009:**
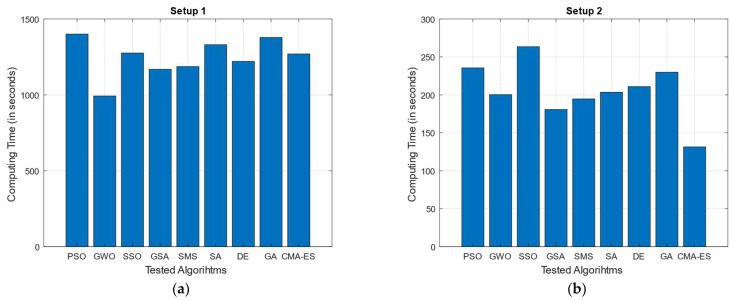
Processing time (in seconds) for all tested MOAs across each of the considered experimental setups. Each algorithm was tested by considering population size of $N_{pop}\;=\;50$ and maximum number function evaluations as $N_{FA}\;=\;5000$ (stop criterion): (**a**) Setup 1; (**b**) Setup 2; (**c**) Setup 3; (**d**) Setup 4.

**Table 1 biomimetics-10-00579-t001:** Proposed experimental layouts.

Layout	Deployment Area Size	Area to Cover	Top View
$L_{1}$	100 m × 100 m	9200 m^2^	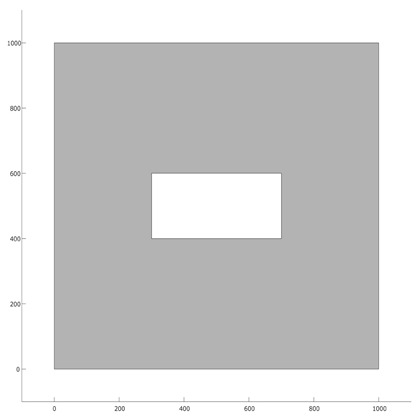
$L_{2}$	100 m × 60 m	4085.5 m^2^	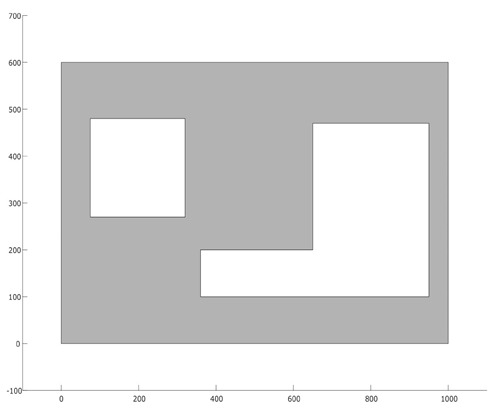

**Table 2 biomimetics-10-00579-t002:** Specifications of image sensors applied for our experiments.

Camera Model	Horizontal Resolution	Maximum Distance for Recognition
$C_{1}$	1280 px	10.24 m
$C_{2}$	3584 px	28.67 m

**Table 3 biomimetics-10-00579-t003:** Experimental setups considered for our experiments.

Experimental Setup	Layout	Camera Model	Max. Number of Sensors ($N_{sensors}$)
Setup 1	$L_{1}$	$C_{1}$	34
Setup 2	$L_{1}$	$C_{2}$	6
Setup 3	$L_{2}$	$C_{1}$	20
Setup 4	$L_{2}$	$C_{2}$	5

**Table 4 biomimetics-10-00579-t004:** Performance-influencing characteristics of the MOAs chosen for our experiments. Interest characteristics include: 1. selection mechanisms, 2. type of attractors (when applied), 3. dependence on iterative process, 4. use of population sorting mechanism, 5. use of measurements related to population/agents, and 6. additional memory requirements.

Tested MOAs	Selection Mechanism	Type of Attractors	Iteration Dependence	Population Sorting Mechanisms	Population Related Measurements	Additional Memory Requirements
PSO	Non-Greedy	Global Best	No	No	No	Yes
GWO	Non-Greedy	Multiple	No	Yes	No	No
SSO	Ind. Greedy	Multiple	No	No	Yes	Yes
GSA	Non-Greedy	Multiple	No	No	Yes	Yes
SMS	Non-Greedy	Global Best	Yes	No	Yes	No
SA	Ind. Greedy	N/A	Yes	No	No	No
DE	Ind. Greedy	N/A	No	No	No	No
GA	Greedy	N/A	No	Yes	No	No
CMA-ES	Non-Greedy	N/A	No	No	No	No

**Table 5 biomimetics-10-00579-t005:** Parameter configurations applied to each MOA included in our experiments.

Algorithm	Parameter Configuration
PSO	The cognitive and social coefficients are set to $c_{1}\;=\;2.0$ and $c_{2}\;=\;2.0$, respectively. Also, the inertia weight factor ω is set to decrease linearly from 0.9 to 0.2 as iterations progress.
GWO	The algorithm’s parameter a is set to decrease linearly from 2.0 to 0.0 as iterations progress.
SSO	The female attraction probability is set as $P_{F}\;=\;0.7$.
GSA	The value for initial gravitation constant is set to $G_{o}\;=\;100$, while the constant parameter alpha is set as α = 20.
SMS	Parameters α, β, H and ρ for each of the algorithm stages (gas, liquid and solid) set as α = [0.8, 0.2, 0.0], β = [0.8, 0.4, 0.1], H = [0.9, 0.2, 0.0] and ρ = [0.8, 1.0; 0.3, 0.6; 0.0, 0.1].
SA	The algorithm’s initial temperature is set to $T^{0}\;=\;1$, while the cooling schedule corresponds to a geometrical cooling scheme with cooling rate β = 0.98.
DE	The crossover rate is set to CR = 0.5, while the differential weight is set as F = 0.2.
GA	The parameter setup for this algorithm is as follows: Crossover probability $c_{p}\;=\;0.7$, mutation probability $m_{p}\;=\;0.3$, crossover range factor γ = 0.4, and mutation rate μ = 0.1.
CMA-ES	The number of parents is set to $\mu \;=\;\frac{N}{2}$(with N denoting the population size). Also, the initial step size is set as $\sigma_{0}\;=\;0.3\left({ub}^{j}\;-\;{lb}^{j}\right)$(where ${lb}^{j}$ and ${ub}^{j}$ stand for the lower and upper function bounds for the j-th decision variable).

**Table 6 biomimetics-10-00579-t006:** Settings for the number of decision variables ($N_{dims}$) and the lower/upper bounds (lb and ub) corresponding to each experimental setup.

Experimental Setup	No. of Decision Variables ($N_{dims}=N_{sensors}\times 2$)	Lower Bounds (lb)	Upper Bounds (ub)
Setup 1	68	[0, 0]	[100, 100]
Setup 2	12	[0, 0]	[100, 100]
Setup 3	40	[0, 0]	[100, 60]
Setup 4	10	[0, 0]	[100, 60]

**Table 7 biomimetics-10-00579-t007:** Performance results of all tested algorithms for Setup 1. The table also shows the ranking (Rank) assigned to each algorithm according to its overall performance.

Setup 1	PSO	GWO	SSO	GSA	SMS	SA	DE	GA	CMA-ES
$f_{mean\;}$	84.6234	77.1047	87.0902	72.0761	52.1583	64.5559	73.6081	91.1077	91.0235
$f_{median}$	84.4077	73.1302	87.2563	72.2618	51.9943	64.0356	73.4302	91.2033	91.1724
$f_{std}$	2.3948	8.6701	1.5441	1.3295	1.8716	3.7935	1.3981	1.402	1.9494
$f_{worst}$	80.1235	68.1901	84.165	69.9594	48.5673	56.9021	71.3179	87.6678	86.5655
$f_{best}$	89.0419	91.6162	89.6659	74.4551	57.0627	72.7558	77.128	93.378	94.2555
**Rank**	4	7	3	6	9	8	5	1	2

**Table 8 biomimetics-10-00579-t008:** Performance results of all tested MOAs for Setup 2. The table also shows the ranking (Rank) assigned to each algorithm according to its overall performance.

Setup 2	PSO	GWO	SSO	GSA	SMS	SA	DE	GA	CMA-ES
$f_{mean\;}$	99.2868	99.291	99.1394	91.4065	24.5198	66.2833	96.5707	99.7542	99.7715
$f_{median}$	99.4223	98.9875	98.9764	91.2894	35.8682	65.6371	96.5295	99.9188	99.9748
$f_{std}$	0.5917	0.4828	0.5449	2.5584	18.7524	8.8946	0.7762	0.2695	0.3499
$f_{worst}$	97.6327	98.7684	98.0602	86.626	0	49.6353	94.7463	99.016	98.8728
$f_{best}$	99.9757	99.946	99.9733	97.9343	46.9373	83.8514	98.1229	99.9738	99.9758
**Rank**	4	3	5	7	9	8	6	2	1

**Table 9 biomimetics-10-00579-t009:** Performance results of all tested MOAs for Setup 3. The table also shows the ranking (Rank) assigned to each algorithm according to its overall performance.

Setup 3	PSO	GWO	SSO	GSA	SMS	SA	DE	GA	CMA-ES
$f_{mean\;}$	90.3033	90.5179	93.18	68.3158	32.6301	53.1749	77.5501	95.1236	81.1087
$f_{median}$	91.1689	91.5495	93.8163	68.3673	32.871	52.698	77.4673	95.3903	82.5378
$f_{std}$	2.7016	5.4455	2.14	2.435	6.5084	7.1647	1.8972	1.4516	5.7681
$f_{worst}$	83.6998	72.0349	87.8428	63.2969	2.0648	33.45	74.6323	92.0142	61.4591
$f_{best}$	95.0913	95.2792	96.3729	73.1362	41.7865	69.504	81.7021	97.5848	89.8897
**Rank**	4	3	2	7	9	8	5	1	6

**Table 10 biomimetics-10-00579-t010:** Performance results of all tested MOAs for Setup 4. The table also shows the ranking (Rank) assigned to each algorithm according to its overall performance.

Setup 4	PSO	GWO	SSO	GSA	SMS	SA	DE	GA	CMA-ES
$f_{mean\;}$	99.1203	97.3447	97.6318	81.1202	7.6064	50.5798	95.899	98.8423	93.58
$f_{median}$	99.1402	98.9898	99.0432	80.8924	6.4786	53.2502	95.7022	99.1104	94.7192
$f_{std}$	0.0582	2.7435	1.9964	3.5418	8.3083	16.5811	1.4433	1.0389	3.9898
$f_{worst}$	99.039	90.3657	92.0773	75.028	0	13.8962	92.3868	94.9475	86.416
$f_{best}$	99.1914	99.1433	99.207	88.5878	24.6234	77.0289	98.0929	99.1774	98.7468
**Rank**	1	4	3	7	9	8	5	2	6

**Table 11 biomimetics-10-00579-t011:** Global rankings of all tested MOAs according to their performance over experimental setups 1 to 4. The table shows the rankings achieved by each algorithm on each experimental setup and their global ranking according to their overall performance.

Exp. Setups	PSO	GWO	SSO	GSA	SMS	SA	DE	GA	CMA-ES
Setup 1	4	7	3	6	9	8	5	1	2
Setup 2	4	3	5	7	9	8	6	2	1
Setup 3	4	3	2	7	9	8	5	1	6
Setup 4	1	4	3	7	9	8	5	2	6
**Overall Ranking**	2	5	3	7	9	8	6	1	4

**Table 12 biomimetics-10-00579-t012:** Processing time (in seconds) of all tested MOAs for each of the considered experimental setups.

Exp. Setup	PSO	GWO	SSO	GSA	SMS	SA	DE	GA	CMA-ES
Setup 1	1400.6557	992.7494	1275.7858	1168.6922	1186.3957	1330.8098	1220.8261	1378.2511	1269.8435
Setup 2	235.5859	200.3457	263.4625	180.7595	194.6946	203.4944	210.9269	229.8822	131.4785
Setup 3	1135.9324	1014.8171	1171.6725	688.6697	814.5918	939.4305	951.1647	1044.7338	820.5319
Setup 4	301.5052	256.6576	355.1201	173.6651	151.724	234.4276	260.6048	288.7302	139.2823

## Data Availability

Raw data supporting the conclusions of this article will be made available by the authors on request.

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
