# Peer review of "Metaheuristics-Assisted Placement of Omnidirectional Image Sensors for Visually Obstructed Environments"

_biomimetics, 2025, doi:10.3390/biomimetics10090579_

Round 1
Reviewer 1 Report
Comments and Suggestions for Authors
This study investigates the performance of various metaheuristic optimization algorithms (MOAs) for optimal camera placement (OCP) of omnidirectional sensors in indoor environments. Two experimental layouts, including deployment areas and obstructions, were tested with different omnidirectional camera models. Algorithms like PSO, GWO, SSO, GSA, SMS, SA, DE, GA, and CMA-ES were evaluated. Results indicate that the success of MOA-based OCP depends heavily on the search strategy, with some methods being more effective than others for this problem.
The paper is of certain practical significance, but there are some issues that need to be addressed:
- The abstract doesn't introduce the significance of the problem, specifically why the study is applied to optimal camera placement (OCP) of omnidirectional image sensors in indoor environments.
- The author's approach seems more like a review, discussing several metaheuristic algorithms applied to OCP, but why not use some mainstream variants for experimentation?
- I believe the experiments are not comprehensive enough; investigating using runtime as a stopping condition would help explore the strengths and weaknesses of different algorithms, ensuring a more thorough comparison.
- The conclusion could analyze future work.
Author Response
COMMENTS 1
The abstract doesn't introduce the significance of the problem, specifically why the study is applied to optimal camera placement (OCP) of omnidirectional image sensors in indoor environments.
RESPONSE 1
The abstract has been slightly modified to introduce the significance of OCP, as well as briefly exposing the presence of some research gaps related to OCP for omnidirectional camera sensors assisted by Metaheuristic Optimization Algorithms.
--
COMMENTS 2
The author's approach seems more like a review, discussing several metaheuristic algorithms applied to OCP, but why not use some mainstream variants for experimentation?
RESPONSE 2
The algorithms applied for our experiments were chosen considering its simplicity (in terms of search strategy and parameter tuning), but more importantly regarding to certain performance influencing characteristics. These are:
- Selection Mechanisms
- Type of attractors (when applied)
- Dependence on iterative process
- Use of population sorting mechanism
- Use of measurements related to population/agents
- Additional memory requirements
A table summarizing (Table 4) these traits was added to the manuscript.
Our selection was made so that none of the tested methods has a common set of these characteristics, as it is of our particular interest to analyze how these collection of traits impact the performance of MOAs over the studied problem (OCP of Image Sensors).
It is worth commenting that part of our planed future work involves testing other more complex MOAs (including mainstream techniques as well as other modified/ hybrid methods).
--
COMMENTS 3
I believe the experiments are not comprehensive enough; investigating using runtime as a stopping condition would help explore the strengths and weaknesses of different algorithms, ensuring a more thorough comparison.
RESPONSE 3
Attending to your recommendations, we have added Table 12 and Figure 9 to the manuscript; these show the average processing time achieved by each MOA (PSO, SSO, GA, etc.) over all proposed experimental setups. In addition, discussion related to these processing times has been added into the Discussion Section (sec. 5).
--
COMMENTS 4
The conclusion could analyze future work.
RESPONSE 4
An additional section exposing some ideas for future work was added after the conclusions section
Reviewer 2 Report
Comments and Suggestions for Authors
The authors present a study on the performance of several popular metaheuristic optimization algorithms when applied to optimal camera placement in indoor environments for omnidirectional imaging. The study examines two experimental setups, which include deployment areas and visual obstructions, as well as two different models of omnidirectional camera. The tested metaheuristics include other evolutionary techniques such as particle swarm optimization, grey wolf optimization, simulated annealing, genetic algorithm, differential evolution, and others. The article is written in a clear and engaging style. The strength of this paper lies in its Section 4, which focuses on experimental results. The authors investigate a complex optimization problem known as Optimal Camera Placement (OCP). This involves deploying omnidirectional image sensors in indoor environments, and carefully designing the distribution of these sensors is essential to ensure maximum coverage. However, the unsupervised nature of non-greedy algorithms makes it more challenging to find optimal solutions, compared to using greedy algorithms. To address this, the authors apply evolutionary algorithms such as Genetic Algorithms (GA) and Particle Swarm Optimization (PSO), among others, and these algorithms demonstrate excellent performance. Despite this, there is room for improvement in the paper:
1. The literature review could be enhanced. It would be beneficial to include more references to works on hybrid evolutionary algorithms, such as CBHPSO, RCGA-PSO, PSO with crossover, PSO with mutation and GA-ACO. Some significant publications that could be cited include:
[1] Akopov A.S. (2024). A Clustering-Based Hybrid Particle Swarm Optimization Algorithm for Solving a Multisectoral Agent-Based Model. Studies in Informatics and Control, 33(2), 83–95. https://doi.org/10.24846/v33i2y202408
[2] Yonggang Chen, Lixiang Li, Jinghua Xiao, Yixian Yang, Jun Liang, Tao Li,
Particle swarm optimizer with crossover operation, Engineering Applications of Artificial Intelligence, Volume 70, 2018, Pages 159-169,
https://doi.org/10.1016/j.engappai.2018.01.009.
2. In section 3. "Metaheuristic optimization algorithms" it is advisable to include a table of comparative analysis of the algorithms considered in the context of the optimization problem considered by the authors. In addition, it is necessary to present a flowchart demonstrating the architecture of the framework, which integrates the optimization problem with evolutionary algorithms such as GA, PSO, GWO, etc. The main goal in OCP is to maximize the area observed by the camera network. However, it is unclear how the model of the search for optimal camera locations connected with evolutionary search methods. The inclusion of flowcharts or pseudocodes could clarify this important aspect.
3. In section 4, "Experimental Results", there is a gap in terms of comparing the time efficiency of algorithms used by authors. Including a small table with metric process times (PT) in section would be useful for the context of solving the problem. The authors found that GA was the overall best-performing algorithm for OCP problems. However, it is not entirely clear how much faster or slower GA is compared to other algorithms when searching for optimal solutions.
Author Response
COMMENTS 1
The literature review could be enhanced. It would be beneficial to include more references to works on hybrid evolutionary algorithms, such as CBHPSO, RCGA-PSO, PSO with crossover, PSO with mutation and GA-ACO. Some significant publications that could be cited include:
[1] Akopov A.S. (2024). A Clustering-Based Hybrid Particle Swarm Optimization Algorithm for Solving a Multisectoral Agent-Based Model. Studies in Informatics and Control, 33(2), 83–95. https://doi.org/10.24846/v33i2y202408
[2] Yonggang Chen, Lixiang Li, Jinghua Xiao, Yixian Yang, Jun Liang, Tao Li,
Particle swarm optimizer with crossover operation, Engineering Applications of Artificial Intelligence, Volume 70, 2018, Pages 159-169,
https://doi.org/10.1016/j.engappai.2018.01.009.
RESPONSE 1
The recommended references, as well as some others related to the topic of modified/hybrid MOAs have been added to the manuscript. Mentions to these works have also been added to section 3.
--
COMMENTS 2
In section 3. "Metaheuristic optimization algorithms" it is advisable to include a table of comparative analysis of the algorithms considered in the context of the optimization problem considered by the authors. In addition, it is necessary to present a flowchart demonstrating the architecture of the framework, which integrates the optimization problem with evolutionary algorithms such as GA, PSO, GWO, etc. The main goal in OCP is to maximize the area observed by the camera network. However, it is unclear how the model of the search for optimal camera locations connected with evolutionary search methods. The inclusion of flowcharts or pseudocodes could clarify this important aspect.
RESPONSE 2
A flowchart (Figure 3) illustrating both, the general process highlighting MOAs’ Main Loop, as well as the Coverage Area Rate Computing Process has been added. Also, an explanation of such a process has been added at the beginning of the Experimental Results section (sec 4.)
--
COMMENTS 3
In section 4, "Experimental Results", there is a gap in terms of comparing the time efficiency of algorithms used by authors. Including a small table with metric process times (PT) in section would be useful for the context of solving the problem. The authors found that GA was the overall best-performing algorithm for OCP problems. However, it is not entirely clear how much faster or slower GA is compared to other algorithms when searching for optimal solutions.
RESPONSE 3
Attending to your recommendations, we have added Table 12 and Figure 9 to the manuscript; these show the average processing time achieved by each MOA (PSO, SSO, GA, etc.) over all proposed experimental setups. In addition, discussion related to these processing times has been added into the Discussion Section (sec. 5).
Round 2
Reviewer 1 Report
Comments and Suggestions for Authors
I don't have any further questions.
Reviewer 2 Report
Comments and Suggestions for Authors
The authors have significantly revised the article in response to reviewers' comments. This study makes a significant contribution to the investigation of potential applications of metaheuristics assisted optimal placement of omnidirectional image sensors in visually obscured environments. Researchers tested a variety of metaheuristic optimization algorithms (MOAs), including popular approaches such as PSO, GWO, SSO, GSA, SMS, SA, DE, GA, and CMA-ES. The authors have obtained experimental results that demonstrate the success of MOA-based optimal camera placement, which heavily depends on the specific search strategies employed by the metaheuristic. Certain approaches have been shown to be more advantageous for solving this problem. This paper provides a compelling example of how metaheuristics can be applied to real-world optimization problems, and I would therefore recommend publishing the work.